# The Immunoprotection of *OmpH* Gene Deletion Mutation of *Pasteurella multocida* on Hemorrhagic Sepsis in Qinghai Yak

**DOI:** 10.3390/vetsci10030221

**Published:** 2023-03-14

**Authors:** Jianlei Jia, Meng Zhao, Kairu Ma, Hongjian Zhang, Linsheng Gui, Huzhi Sun, Huiying Ren, Tamaki Okabayashi, Jing Zhao

**Affiliations:** 1College of Agriculture and Animal Husbandry, Qinghai University, Xining 810016, China; 2School of Life Sciences, Qilu Normal University, Jinan 250200, China; 3Qingdao Phagepharm Bio-Tech Co., Ltd., Qingdao 266109, China; 4College of Veterinary Medicine, Qingdao Agricultural University, Qingdao 266109, China; 5Department of Veterinary Science, Faculty of Agriculture, University of Miyazaki, Miyazaki 889-2192, Japan

**Keywords:** Qinghai yak, *OmpH*, hemorrhagic sepsis, immunoprotection, proteomics

## Abstract

**Simple Summary:**

*Pasteurella multocida* is a pathogen that seriously harms the cattle breeding industry. In this study, the clinical signs, bacterial load and proteomics of infected yaks were studied by infecting yaks with wild-type (WT) (P0901) and OmpH-deficient strains (ΔOmpH) of *Pasteurella multocida*. We found that compared with the mutant strain, the titer of wild-type strains was significantly higher in tissues. Additionally, compared with other organs, the bacteria titer was significantly higher in the spleen. Compared with the WT P0910 strain, the mutant strain generated milder pathological changes in the tissues of yak. Proteomics analysis revealed that 57 of the 773 proteins expressed in *Pasteurella multocida* were significantly differentially expressed between the ΔOmpH and P0910 groups. It was also found that P0910 and ΔOmpH in *Pasteurella multocida* infection activated the expression of ropE, HSPBP1, FERH, ATP10A, ABCA13, RRP7A, IL-10, IFN-γ, IL-17A, EGFR and dnaJ. Overall, deletion of the *OmpH* gene weakened the virulence but maintained the immunogenicity of *Pasteurella multocida* in yak. The findings of this study provide a strong foundation for the pathogenesis of *Pasteurella multocida* and the management of related septicemia in yaks.

**Abstract:**

*OmpH* is among the most important virulence factors of *Pasteurella multocida*, which mediates septicemia in yaks (*Bos grunniens* I) after infection with the bacteria. In the present study, yaks were infected with wild-type (WT) (P0910) and OmpH-deficient (ΔOmpH) *P. multocida* strains. The mutant strain was generated through the reverse genetic operation system of pathogens and proteomics technology. The live-cell bacterial count and clinical manifestations of *P. multocida* infection in Qinghai yak tissues (thymus, lung, spleen, lymph node, liver, kidney, and heart) were analyzed. The expression of differential proteins in the yak spleen under different treatments was analyzed using the marker-free method. We found that compared with the mutant strain, the titer of wild-type strains was significantly higher in tissues. Additionally, compared with other organs, the bacteria titer was significantly higher in the spleen. Compared with the WT p0910 strain, the mutant strain generated milder pathological changes in the tissues of yak. Proteomics analysis revealed that 57 of the 773 proteins expressed in *P. multocida* were significantly differentially expressed between the ΔOmpH and P0910 groups. Of the 57, 14 were over-expressed, whereas 43 were under-expressed. The differentially expressed proteins in the ΔompH group regulated the ABC transporter (ATP-powered translocation of many substrates across membranes) system, the two-component system, RNA degradation, RNA transcription, glycolysis/gluconeogenesis, biosynthesis of ubiquinone and other terpenoid-quinones, oxidative phosphorylation (citrate cycle) as well as fructose and mannose metabolism. The relationship among 54 significantly regulated proteins was analyzed using STRING. We found that WT P0910 and ΔOmpH of *P. multocida* infection activated the expression of ropE, HSPBP1, FERH, ATP10A, ABCA13, RRP7A, IL-10, IFN-γ, IL-17A, EGFR, and dnaJ. Overall, deletion of the *OmpH* gene weakened the virulence but maintained the immunogenicity of *P. multocida* in yak. The findings of this study provide a strong foundation for the pathogenesis of *P. multocida* and the management of related septicemia in yaks.

## 1. Introduction

Yak (*Bos grunniens*) is the main bovine on the Tibetan plateau and thrives well at an altitude above 3000 m. Yak is a versatile and hardy animal that thrives in very harsh environments [1]. The first domestication of the animal dates back to several thousand years ago [2]. Yak production has been the main source of income for the local residents of the Tibetan plateau. Additionally, yak hemorrhagic septicemia is an important disease affecting the yak industry. Yak hemorrhage septicemia is an infectious bacterial disease caused by *Pasteurella multocida* (*P. multocida*), characterized by high fever, pneumonia, occasional acute gastroenteritis, and internal organ bleeding. It is among the most important diseases affecting Qinghai yak production [3,4]. Yaks suffer an acute form of this disease, and the infection develops rapidly; further, the animal dies before diagnosis. Besides the substantial economic losses caused by hemorrhage septicemia, *P. multocida* poses a substantial public health risk. However, the genetic blueprint and the virulence and immune response mechanism against *P. multocida* in yak are unclear.

*Pasteurella multocida* is a pathogen with multivirulence factors, including the capsule, lipopolysaccharide, toxin, and the outer membrane protein [5]. However, the pathogenesis of *P. multocida* is mainly mediated by virulence factors that adhere and invade the host tissue cells [6]. Apart from pathogenicity, virulence factors are critical in the survival and reproduction of the pathogen in the host. The *outer membrane protein H (OmpH)*, among the most important outer membrane proteins of *P. multocida*, plays a key role in the pathogenicity and invasiveness of the bacteria [7]. The *OmpH* gene Pasteurella multicida of poultry origin is a 1062 bp fragment and displays 99.9% homology with the cp39 gene that codes for the capsule protein [8,9]. *OmpH* gene deletion significantly reduces the resistance of the mutant bacteria to phagocytosis. Compared with the live attenuated bacteria, inactivated *P. multocida* display stronger immunogenicity and adhesion to the HeLa cells. The 39 kDa protein encoded by the *OmpH* gene is related to the pathogenicity of *P. multocida*. Interestingly, the 39 kDa protein in the *P. multocida* capsule is present in the inactivated but not live bacteria. The *OmpH* mutant provides 100% protection (against *P. multocide*) in poultry [10,11]. Previous studies have found no significant correlation between altitude and the incidence of hemorrhagic septicemia in yaks. These findings suggest that the pathogenicity of *P. multocida* in Qinghai yaks might be related to hypoxia [12].

*Pasteurella multocida* has been detected in tissues, secretions, and excreta of livestock and poultry. The bacteria have also been detected in the tonsils and upper respiratory tract of healthy livestock and poultry [13]. *Pasteurella multocida* infection mainly occurs via the digestive and respiratory routes, and transmission through animal bites, damaged skin, mucous membrane, insect vectors, contaminated equipment, drinking water, and feeds has also been reported [5,14]. Whether *OmpH* participates in the adhesion, invasion, pathogenesis, and virulence of *P. multocida* in yak is unclear. We explored this phenomenon in the present study. The effect of *OmpH* deletion on the immunogenicity of *P. multocida* was also explored. Reverse genetic operation systems of pathogens and proteomics technology have become powerful tools for assessing the effect of specific genes deletion on the virulence and pathogenicity of pathogenic organisms. Herein, Qinghai yaks were infected with wild-type strain (P0910) and OmpH-deficient strains (ΔOmpH). The corresponding viable bacteria count in tissues of this animal (thymus, lung, spleen, lymph node, liver, kidney, and heart) was then quantified. The subsequent clinical manifestations in these organs and the immune response to these pathogens in the yak spleen were examined using label-free mass spectrometry. Herein, we explored the effect of *OmpH* gene deletion on the virulence and immunogenicity of *P. multocida* in yak. The findings of this study provide a solid background on the infection, pathogenicity, immunogenicity, as well as control and management of *P. multocida* infection, improving yak production.

## 2. Materials and Methods

### 2.1. Ethics Statement

The protocol for the animal experiments was approved by the Animal Care Committee of Qinghai University, China. Slaughtering of the animals was performed in accordance with the National Administration of Cattle Slaughtering and Quarantine guidelines (Qinghai, China, 20170312-07).

### 2.2. Study Location and Animals

This research was conducted at Haibei Tibetan Autonomous Prefecture (HTAP), Qinghai Province, China. The region is located at more than 3000 m above sea level, northeastern of Qinghai–Tibetan Plateau. A total of 15 Qinghai yak calves (0.5 years old, half male and half female) were used in this study and were fed through grazing on natural pastures.

### 2.3. Bacterial Strains, Plasmid and Culture Condition

The *P. multocida* P0910 strain used in this study was isolated from dead yaks in Qilian County, Qinghai Province. *Pasteurella multocida* was cultured in Trypticase Soy Broth (TSB) at 37 °C under constant shaking and the concentration of the bacterial solution was determined by plate counting. *E. coli* DH5α competent cells were purchased from Sangon (Sangon Biotech, Shanghai, China). *E. coli* DH5α recombinant strains were screened on plates of Luria–Bertani (LB) medium supplemented with 100 μg/mL ampicillin. The pEX18AP plasmid was obtained from Dr Qingmin Wu, China Agricultural University.

### 2.4. Construction of Mutant Strain

Primers used in mutant strain construction are listed in Table 1. The PCR amplified a 362 bp homologous recombination arm upstream by Wu2309 and Wu2310. The 462 bp homologous recombinant arm of the *OmpH* gene downstream was amplified by Wu2011 and Wu2012. The upper and lower arm fragments were ligated by overlapping PCR and inserted into the EcoR I and BamH I sites of pEX18AP using the In-Fusion^®^HD Cloning Kit (Takara, Kusatsu, Japan) to generate the recombinant plasmid pEX18AP-OMPH. The recombinant plasmid pEX18AP-OMPH was transformed into *E. coli* DH5α and positive samples were screened using LB agar with resistance. The pEX18AP-OMPH plasmid was transferred into P0910 receptor cells by electroporation, and the ΔOmpH mutant was screened on TSB ager with 100 μg/mL ampicillin and verified by PCR.

### 2.5. Vaccination Experiments in Yaks for Protection

The calves were randomly divided into three groups. Six yak calves in group A received a subcutaneous injection of P0910 (2.0 × 10^6^ CFU/mL). Another six yaks in group B received a subcutaneous injection of ΔOmpH (2.0 × 10^6^ CFU/mL, Group B). Three yaks in the check group (Group C) received a subcutaneous injection of sterilized normal saline. After P0910 and ΔOmpH were adjusted to 2.0 × 10^6^ CFU/mL, 1 mL of different bacterial solution was injected subcutaneously into different groups of yaks. Yaks in each group were raised separately. The morbidity and mortality in each group were observed every day. The thymus, lung, spleen, lymph node, liver, kidney, and heart tissues from the animals were collected at 24 and 48 h. The yaks that did not die at 24 and 48 h were dissected and the pathological changes were observed.

### 2.6. Bacterial Loads in Different Tissues

Lesions and viable bacteria count in yaks that died naturally or were sacrificed were observed and quantified, respectively. The thymus, lung, spleen, lymph node, liver, kidney, and heart tissues from the animals stored 1.5 mL centrifugal tubes. The bacteria count was estimated based on 10 serial dilutions (10 μL for each dilution) of the tissue samples using the micro-inoculation colony counting method, each sample was repeated 3 times.

### 2.7. Proteomic Analysis

The thymus, lung, spleen, lymph node, liver, kidney, and heart tissues from the animals for proteomics analyses were stored in 1.5 mL centrifugal tubes. Surface contaminants were washed off using ultracold (−80 °C) phosphate-buffered saline solution. Proteomics analysis was performed at Novogene (China, Beijing). The experiments were performed indecently three times. Proteins in the spleen were extracted using lysis buffer. The proteins (250 μg of each sample) were digested using the FASP procedure [15]. The amount of different proteins in the tissues samples was quantified using a Q Exactive mass spectrometer equipped with the Easy nLC system (Thermo Fisher Scientific, MA, USA). The resultant data were analyzed using MaxQuant software based on the UniProt Bos taurus database. Real-time quantitative fluorescence PCR was used to analyze the copy number of 8 differentially expressed protein coding genes (*FERH*, *HSPBP1*, *ABCA13 ATP10A RRP7A, IL-10*, *IFN-γ* and *dnaJ*). All the primers were synthesized and provided by Sangon (Sangon Bio-tech, Shanghai, China) (Table 1).

### 2.8. Statistical Analysis

Bioinformatics analyses were performed as previously described by Jia [9]. Functional annotation and classification of all the identified proteins were performed using the Blast2GO program, based on the data in the uniport database. Pathways regulated by the differently expressed genes (DEGs) were identified using the KEGG analysis http://www.genome.jp/kegg/mapper.html (accessed on 15 November 2021). Significantly regulated pathways were identified using the Fisher’s exact test with statistical significance set at *p* < 0.05. The protein–protein interactions (PPIs) were predicted using the STRING program (http://string-db.org/ (accessed on 15 November 2021)). 

Continuous normally distributed data were expressed as the mean ± standard deviation (SD). Differences between groups were analyzed using the Duncan’s post hoc test. Statistical significance was set at *p* < 0.05.

## 3. Results

### 3.1. Construction of Mutant Strain

To study the role of the *OmpH* gene in *P. multocida*, we constructed *OmpH* deletion strain (ΔOmpH) of *P. multocida* P0910. PCR results showed that the upstream and downstream homologous arms (362 bp and 462 bp) of the *OmpH* gene were amplified (Figure 1a). They were connected to plasmid pEX18AP to form recombinant plasmid pEX18AP-OMPH (Figure 1b). In the strain ΔOmpH, only the upstream and downstream homologous arms of the *OmpH* gene can be amplified, but the *OmpH* gene cannot be amplified (Figure 1c).

### 3.2. Dynamics of the Proliferation of P0910 and ΔOmpH of P. multocida

The pathogenicity experiment revealed that the viable bacterial counts of group A and group B (ΔOmpH) in tissues and organs strongly depended on the infection time and strains (*p* < 0.05), but there was no interaction between the two factors (*p* > 0.05). Additionally, the viable count was significantly high in group A (P0910), relative to group B (ΔOmpH) (*p* < 0.05). Just like pathogenicity, infection time greatly influenced the number of viable bacterial count (*p* < 0.05), but there was no significant interaction between organs and infection time and the number of viable bacterial count (*p* > 0.05). Compared with other organs, the viable bacterial count was significantly higher in the spleen (*p* < 0.05) because the spleen is probably the main target organ for the bacteria (Table 2).

### 3.3. Clinical Symptoms of P0910 and ΔOmpH of P. multocida Infection

For P0910, the yaks developed a fever after about 11 h (Figure 2a), gradually followed by anorexia and dyspnea, with loud wheezing, a dry nose, and depression appearing in severe cases. For the ΔOmpH infection, symptoms appeared after about 16 h (Figure 2b).

After 24 h infection with P0910, mild bleeding was detected in the thymus, but no visible pathological changes were observed in the lung, heart, liver, spleen, and kidney (Figure 2c). After 48 h (Figure 2d), hemorrhage occurred in the thymus and endo-epicardial tissues followed by pericardial infusion and swelling and hardening of the liver, spleen, and kidney. Petechial hemorrhage was also observed in these organs. For the ΔOmpH infection group, the typical pathological changes were not apparent between 24 and 48 h and were milder than in the P0910 infection group.

### 3.4. Proteomics Analysis

#### 3.4.1. Proteome Profiles during Infection with *P. multocida* P0910 and ΔOmpH

A total of 773 proteins were differentially expressed between the P0910 and ΔOmpH *P. multocida* infection groups (*p* < 0.05; quantitative ratio of >2 or <0.5). Additionally, compared with the P0910 group, 439 and 334 proteins were over and under-expressed, respectively, in the ΔOmpH infection group. Among them, 57 were significantly differently expressed, in which 14 were significantly over-expressed, whereas 43 proteins were significantly under-expressed (Figure 3).

#### 3.4.2. Bioinformatics Analyses

The cellular components (CC), molecular function (MF), and biological process (BP) regulated by the significantly regulated proteins between the P0910 and ΔOmpH groups were evaluated using Gene Ontology (GO) analysis (Figure 4a). The major significantly regulated proteins were membrane (GO:0016020), membrane part (GO:0044425), intrinsic component of membrane (GO:0031224), peripheral cell proteins (GO:0071944), integral membrane proteins (GO:0016021), and plasma membrane protein for cellular components (GO:0005886). The proteins regulated several biological functions including hydrolase activity (GO:0016787), ATPase activity (GO:0042623), small molecule binding (GO:0036094), transporter activity (GO:0005215), carbohydrate transportation (GO:1901476), transmembrane carbohydrate transportation (GO:0015144), transmembrane sugar transportation (GO:0051119), transmembrane metal ions transportation (GO:0046915), transmembrane inorganic cation transportation (GO:0022890), substrate-specific transmembrane transportation (GO:0022891), P-P-bond-hydrolysis-driven transmembrane transportation (GO:0015405), primary active transmembrane transportation (GO:0015399), active transmembrane transport activity (GO:0022804), transmembrane transport activity (GO:0022857), and electron carrier activity (GO:0009055), proteolysis (GO:0006508), protein catabolism (GO:0030163), metabolism of macromolecules (GO:0015145), regulation of primary metabolic processes (GO:0080090), metabolism of organic substances (GO:1901575), and response to stimuli (GO:0050896).

KEGG pathway enrichment analysis revealed that 20 major pathways were regulated by the 57 differentially expressed proteins between the P0910 and ΔOmpH groups (Figure 4b). The most significant ones included the ABC transportation (ko02010), two-component system (ko02020), RNA degradation (ko03018), RNA polymerase (ko03020), glycolysis/gluconeogenesis (ko00010), ubiquinone and another terpenoid-quinone biosynthesis (ko00130), the citrate cycle (TCA cycle) (ko00020) as well as fructose and mannose metabolism (ko00051) pathways. KEGG enrichment pathway and GO analysis results revealed comparable findings.

Further analyses of the 57 differently expressed proteins between the ΔOmpH and the P0910 groups revealed that coding (ropE, HSPBP1, FERH, ATP10A, and ABCA13) and immune response-related proteins (RRP7A, IL-10, IFN-γ, IL-17A, EGFR, and dnaJ) formed a complex and intricate protein interaction network (Figure 4c).

#### 3.4.3. Validation of the Differentially Expressed Proteins Using qPCR

Real-time quantitative fluorescence PCR was used to analyze the copy number of 8 differentially expressed protein coding genes (*FERH*, *HSPBP1*, *ABCA13 ATP10A RRP7A*, *IL-10*, *IFN-γ* and *dnaJ*). We found that the gene expression of *FERH, HSPBP1*, and *ABCA13* genes was significantly higher in the ΔOmpH group, relative to the P0910 group (*p* < 0.05), but there was no significant difference in the expression of ATP10A between the two groups. However, the expressions of RRP7A, IL-10, IFN-γ, and dnaJ were (significantly high) higher in the P0910 than in the ΔOmpH group (Figure 5; *p* < 0.05). Overall, qPCR and the label-free analysis revealed comparable findings regarding the selected differentially expressed proteins (Figure 5).

## 4. Discussion

With the progress of biotechnology, especially the deepening of contemporary bacterial molecular genetics and molecular biology, through gene recombination, gene insertion, gene deletion and other technologies can greatly promote the research and development of molecular pathogenic mechanism of pathogenic bacteria and high-tech products such as vaccines and diagnostic reagents [16,17,18,19]. With the complete gene sequence of *P. multocida* decoded, it is possible to analyze and study the gene function of Pasteurella using this techniq. In this study, *OmpH* gene deletion was selected as the deletion target and pEX18AP plasmid with positive Amp resistance gene. It was used as deletion vector to construct unlabeled *ompH* gene deletion strains of *P. multocida* successfully. No new genes were inserted in the construction process to avoid the interference of foreign genes on the missing strains and potential biosafety risks. Proteomics analysis was used to understand the differences in bacterial protein expression under different conditions, and to provide evidence for understanding the pathogenic mechanism of host and environment. The differential proteins between the parent strain and the deletion mutant were detected to further study the interaction of multiple *P. multocida* virulence factors.

*OmpH* is among the most important virulence factors for *P. multocida* [7]. It mediates attachment and penetration of the bacteria into the host cells and the resultant pathogenesis, survival, and reproduction of *P. multocida* in the host [19]. Recent proteomic reports have revealed that the *OmpH* gene mediates the pathogenicity of *P. multocida*. Deletion of this gene reduced hemorrhagic sepsis following *P. multocida* infection [20]. The pathogenesis of *P. multocida* to the hosts is due to many virulence factors, including the bacterial capsule [21,22]. Non-capsulated strains were often isolated from chronic cases and showed low virulence, but were still observed to be pathogenic to the natural hosts [23]. It has been shown that deletion of the *OmpH* gene of *P. multocida* is capable of destroying the bacterial capsule. The deletion of the *OmpH* gene resulted in a significant reduction in toxicity and the deletion strain provided a good protection against both chickens and mice [9]. This is consistent with our study—by injecting WTP0910 and ΔompH in Qinghai yaks, we found that the Qinghai yaks injected with ΔompH at the minimum lethal dose of P0910 were able to survive. Additionally, comparing the bacterial load of different organs at different injection times, the bacterium of ΔompH was smaller than P0910. *P. multocida* has been detected in tissues, secretions, body fluids, feces of sick livestock and poultry, and in tonsils (lymphoid tissues in the throat) and respiratory tract of some healthy livestock and poultry. Upon entry into the body, *P. multocida* evades phagocytosis of macrophages to colonize the lungs [24]. Macrophages eliminate pathogenic microorganisms through phagocytosis and apoptosis [25]. Interestingly, injection with anti-OmpH antibodies inhibited the phagocytosis of (the mutant strain but enhanced that of wild-type strain) the wild-type strains and complementary strains were enhanced. In related research, Lei injected mice with the wild-type strain and, later, anti-OmpH antibodies [18]. They found that injection of anti-OmpH antibodies enhanced the phagocytosis of *P. multocida* wild-type strain by macrophages. This implies that inhibition of OmpH activity promotes the phagocytic activity of macrophages against the p0910 wild-type strain. 

*Pasteurella multocida* is an economically important pathogen of animals. Proteomics analyses revealed had been a key method to clarify the pathogenesis of Pasteurellosis [26]. Furthermore, we can understand the protein expression differences of bacteria under different conditions to provide the basis for understanding the pathogenic mechanism of host and environment. the differential proteins between parental strains and deletion mutation strains were detected to further study the interaction of virulence factors of *P. multocida*, and the pathogenic mechanism of *P. multocida* and the immunoprotection of *OmpH* gene deletion mutation strains were analyzed. The yak spleen proteome with treatment by P0910 and ΔompH of *P. multocida* were screened and analyzed by label-free mass spectrometry. Results showed that of the 773 proteins (expressed in wild-type P0910 *P. multocida*), 57 were significantly differentially expressed in ΔOmpH mutant strains. Of the 57, 14 were over-expressed, whereas 43 of them were under-expressed in the ΔOmpH strain. The differentially expressed proteins in ΔOmpH strain regulated ABC transportation, two-component system, RNA degradation, RNA polymerase, glycolysis/gluconeogenesis, ubiquinone and other terpenoid-quinone biosynthesis, the citrate cycle (TCA cycle) as well as fructose and mannose metabolism. The expression of ABC transporters, the activities of the two-component system, RNA degradation, and RNA transcription were significantly high in the ΔOmpH group. In contrast, glycolysis/gluconeogenesis, biosynthesis of ubiquinone, and other terpenoid-quinones, the citrate cycle (TCA cycle), and fructose and mannose metabolism were significantly low in the ΔOmpH group. STRING further revealed a complex and intricate connection among 54 of the 57 differently expressed proteins. Injection of wild-type P0910 and ΔOmpH mutant *P. multocida* strains activated the activities of ropE, HSPBP1, FERH, ATP10A, ABCA13, RRP7A, IL-10, IFN-γ, IL-17A, EGFR, and dnaJ. Thus, these proteins are potential markers for hemorrhagic sepsis in Qinghai yak following *P. multocida* and could also explain the molecular mechanism underlying the pathogenesis of *P. multocida.*

*Pasteurella multocida* infection disrupts pathways that participate in the expression of several key proteins (ABC transportation, two-component system, RNA degradation, and RNA transcription), and immune response-related (glycolysis/gluconeogenesis, ubiquinone, and biosynthesis of other terpenoid-s, the citrate cycle (TCA cycle) as well as Fructos fructose e and mannose metabolism) pathways [27,28]. PPI network analysis revealed a complex and intricate interaction among the aforementioned pathways and related proteins. ABC transporter proteins actively translocate from the periplasm to the cytoplasm [29,30]. RpoE is a sigma factor that mediates the expression of proteins that constitutes the periplasmic and outer membrane of (cells and cellular macromolecules) components [31]. Additionally, it regulates responses to bacterial stress. RpoE is among the virulence proteins in *P. multocida* and participates in maintaining bacterial growth and survival in adverse environments. Heat shock proteins (HSPs) are highly conserved in eukaryotes and prokaryotes and are essential for the survival of organisms [32]. DNA J is among the most important members of the HSP family. It is a molecular chaperone that stimulates ATPase and folding of HSP70 DNA K. DNA J is a virulence factor critical to the pathogenesis of bacteria such as *Edwardsiella tarda* [33]. Mucosal immunization with DNA J induces mucosal immunity and stimulates the secretion of IL-10 and IFN-γ and IL-17A, which reduces colonization of pneumococcal bacteria in the nose and lung. DNA J may be a chaperone protein of DNA K and delays E. tarda infection in fish. FERH is a nuclear protein (essential) necessary for cell growth. Inhibition of FERH expression impairs normal spindle formation and arrests the cell cycle at M-phase. Additionally, FERH regulates signal transduction, intracellular signaling cascade, and glucose and lipid metabolism. rrp7a proteins regulate immune response, the proliferation of blood cells [34]. ATP10A is a transmembrane protein in the p-type cation transport ATPase family [15]. Abca13 is a membrane protein that facilitates numerous substances across the cell membrane. Given that it participates in maintaining an optimal supply of nutrients and excretion of waste material, it may be an important virulence factor in *P. multocida*.

## 5. Conclusions

Deletion of the *OmpH* gene disrupts the normal expression of specific proteins in *P. multocida*. Additionally, infection with *OmpH* mutant *P. multocida* disrupts the expression of several proteins in the yak spleen tissue, affecting immune response against the bacteria. *OmpH* gene deletion influences the pathogenesis of *P. multocida* in Qinghai yak, including the associated hemorrhagic sepsis. These findings deepen our understanding of the pathogenesis and control of *P. multocida* in Qinghai yak.

## Figures and Tables

**Figure 1 vetsci-10-00221-f001:**
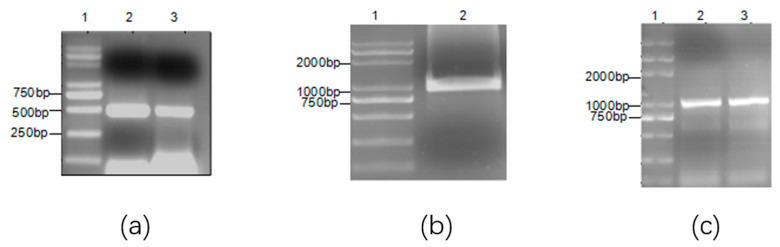
Construction of recombinant plasmid and deletion strain. (**a**) Amplification of upstream and downstream homologous arms. 1, Marker, 2, upstreamhomologous arms, 3, downstream homologous arms. (**b**) Amplification of recombinant plasmid pEX18AP-OMPH. 1, Marker, 2, upstream and downstream homologous arms. (**c**) Amplification of the *OmpH* gene in the strain ΔOmpH. 1, Marker, 2,3, the *OmpH* gene in the strain ΔOmpH.

**Figure 2 vetsci-10-00221-f002:**
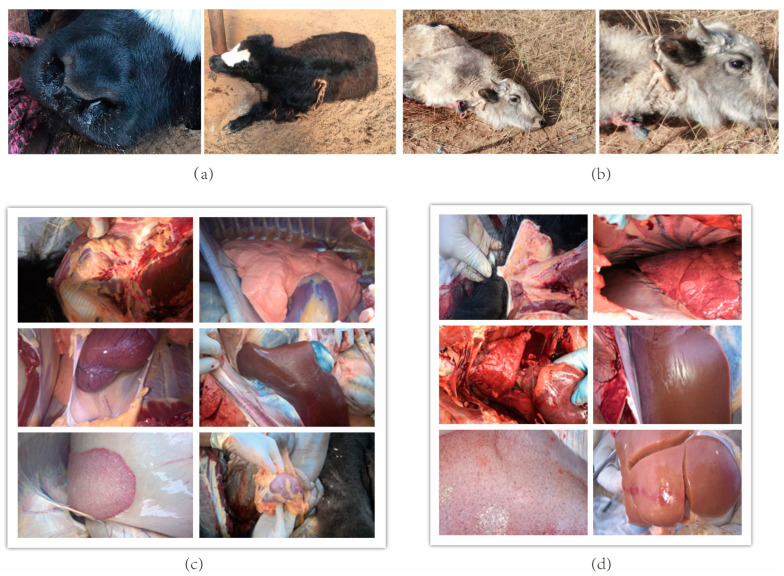
Clinical symptoms of P0910 and ΔOmpH of *P. multocida* infection. (**a**) Clinical symptoms of P0910 after 11 h (**b**) Clinical symptoms of ΔOmpH after 16 h (**c**). Pathological changes after 24 h (**d**) Pathological changes after 48 h.

**Figure 3 vetsci-10-00221-f003:**
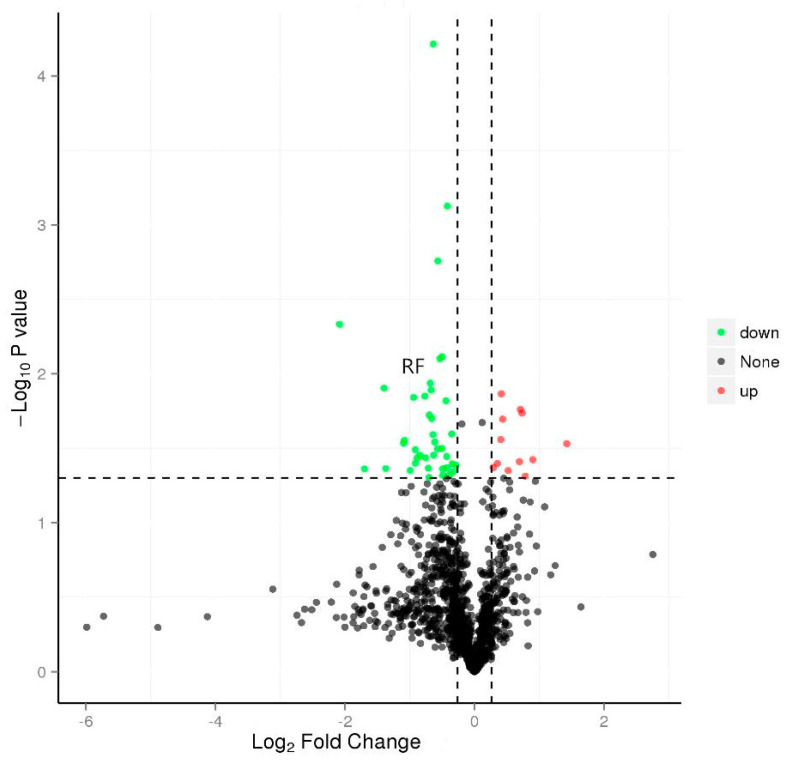
Differentially expressed proteins (DEPs) of cluster evaluation.

**Figure 4 vetsci-10-00221-f004:**
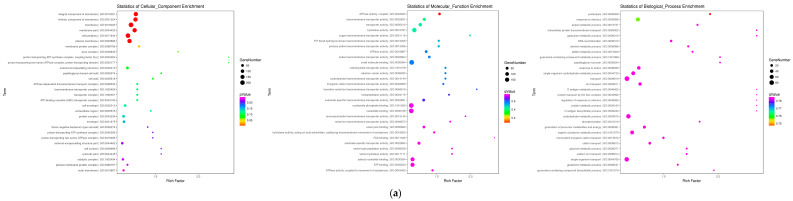
Differentially expressed proteins (DEPs) of different evaluation. (**a**) Differentially expressed proteins (DEPs) of GO evaluation (**b**) Differentially expressed proteins (DEPs) of KEGG evaluation (**c**) Differentially expressed proteins (DEPs) of PPI evaluation.

**Figure 5 vetsci-10-00221-f005:**
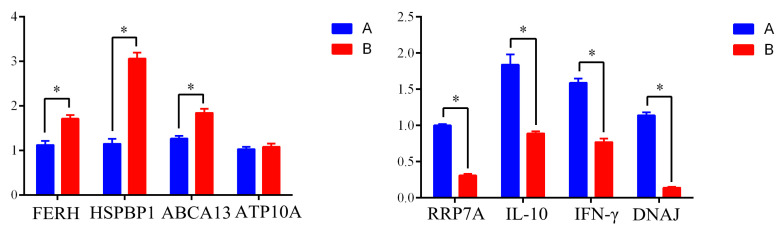
Comparison of 8 differentially expressed protein coding genes (*FERH*, *HSPBP1*, *ABCA13 ATP10A RRP7A*, *IL-10*, *IFN-γ* and *dnaJ*) in Yaks artificially infected with P0910 (A) and ΔompH (B). Values carrying asterisk (*) are statistically different at *p* < 0.05.

**Table 1 vetsci-10-00221-t001:** The primers in this study.

Name	Sequence (5′-3′)
Wu2309	5′ GGAATTCAATAGAGGCATTTACCCG 3′
Wu2310	5′ GGTACC CATTTACATCAACTTTTG 3′
Wu2311	5′TGTAAATGGGTACCTGTTGAAGGTGGCTGG 3′
Wu2312	5′ CGGGATCC AACGCACTTCAACTTGTCC 3′
RRP7A-F	5′ CCCCAAACCAGTTCCTG 3′
RRP7A-R	5′ GCACCGAGTCCGTGTAAT 3′
ABCA13-F	5′ CTGAACGAGGACAAGAT 3′
ABCA13-R	5′ AGACGGAGACCAAGTAA 3′
ATP10A-F	5′ TTTCAGTCCGCCATTC 3′
ATP10A-R	5′ CGGTGAGAACCCAAATC 3′
IFN-γ-F	5′ ATGACACCACCTGAACGTCTCTTC 3′
IFN-γ-R	5′ CTACAGAGCGAAGGCTCCAAAGAAGACAGTACT 3′
IL-10-F	5′ AGGGCACCCAGTCTGAGAACA 3′
IL-10-R	5′ CGGCCTTGCTCTTGTTTTCAC 3′
dnaJ-F	5′ TGTTGCCTACGATACCCTAAGC 3′
dnaJ-R	5′ GCTTGCTCACCGCCTGTAA 3′
HSPBP1-F	5′ CTTTGCTCCATGGGGATGGT 3′
HSPBP1-R	5′ GATCCATGCTGTCGTCGGTA 3′

**Table 2 vetsci-10-00221-t002:** The viable bacterial counts of *P. multocida* in various organs of yaks after artificial infection with P0910 and ΔOmpH.

Post-Infection Time/h	Number	Thymus/CFU/g	Lungs/CFU/g	Spleen/CFU/g	Lymph Nodes/CFU/g	Liver/CFU/g	Kidneys/CFU/g	Heart/CFU/g
24	P0910-1	3.0 × 10^6^	2.67 × 10^4^	3.57 × 10^5^	1.23 × 10^5^	6.67 × 10^4^	2.43 × 10^4^	1.33 × 10^3^
	P0910-2	1.23 × 10^6^	1.98 × 10^4^	7.33 × 10^4^	1.60 × 10^4^	0.67 × 10^4^	1.43 × 10^4^	1.21 × 10^3^
	P0910-3	5.66 × 10^5^	0.91 × 10^3^	2.67 × 10^3^	2.31 × 10^3^	0.16 × 10^3^	4.28 × 10^2^	0.68 × 10^2^
	ΔOmpH-1	3.7 × 10^5^	1.02 × 10^3^	1.59 × 10^4^	3.28 × 10^4^	3.66 × 10^3^	4.48 × 10^3^	2.00 × 10^2^
	ΔOmpH-2	2.3 × 10^4^	2.36 × 10^4^	3.33 × 10^2^	2.77 × 10^3^	5.67 × 10^2^	2.97 × 10^3^	4.29 × 10^3^
	ΔOmpH-3	4.8 × 10^5^	6.28 × 10^3^	8.16 × 10^4^	6.31 × 10^2^	4.16 × 10^4^	3.28 × 10^3^	1.87 × 10^2^
48	P0910-4	1.98 × 10^7^	3.12 × 10^7^	2.33 × 10^6^	9.43 × 10^6^	1.06 × 10^7^	8.93 × 10^5^	6.67 × 10^5^
	P0910-5	1.89 × 10^7^	1.50 × 10^5^	6.98 × 10^4^	4.13 × 10^4^	1.70 × 10^5^	7.93 × 10^4^	4.71 × 10^4^
	P0910-6	1.43 × 10^6^	0.88 × 10^3^	4.11 × 10^4^	2.23 × 10^4^	3.18 × 10^3^	6.26 × 10^3^	0.91 × 10^3^
	ΔOmpH-4	1.3 × 10^6^	4.62 × 10^4^	5.39 × 10^5^	4.09 × 10^3^	5.58 × 10^5^	5.13 × 10^4^	9.11 × 10^3^
	ΔOmpH-5	7.21 × 10^6^	5.59 × 10^6^	4.18 × 10^4^	6.12 × 10^3^	7.70 × 10^4^	8.85 × 10^3^	6.68 × 10^3^
	ΔOmpH-6	5.00 × 10^6^	5.80 × 10^3^	2.15 × 10^4^	7.55 × 10^4^	9.22 × 10^3^	7.52 × 10^3^	3.04 × 10^2^

## Data Availability

The datasets generated during the current study are available from the corresponding author upon reasonable request.

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
