# Peer review of "The Immunoprotection of OmpH Gene Deletion Mutation of Pasteurella multocida on Hemorrhagic Sepsis in Qinghai Yak"

_vetsci, 2023, doi:10.3390/vetsci10030221_

Round 1

Author Response

We have attached our reply. Please check the attachment.

Reviewer 2 Report

P. multocida is the main pathogen of yak and pose great threat to yak health. Thus, it is urgently needed to develop valid vaccine against P. multocida infection in yak. The purpose of this study was to determine whether outer membrane protein OmpH participates in the adhesion, invasion, pathogenesis, and virulence of P. multocida in yak, and to explore the effect of OmpH deletion on the immunogenicity of P. multocida. This is a meaningful research. However, I think it is important to add a group to evaluate the prevention of ompH deletion mutant against P. multocida infection, and the author should pay more attention to the pathogenesis of OmpH-deficient strains(ΔOmpH) and the protective effect ofΔOmpH against P. multocida infection, but not the proteomics. In addition, the manuscript is not well organized, it should be rewritten to improve its logic and fluency, especially the Instruction and the Discussion sections. The question and the purpose of this study should be logically deduced from the background of P. multocida and Yak.

Other comments:

Line 56-57, I think that between the two sentences : “Yak production has been the main source of income for the local residents  of  the  Tibetan  plateau.” and “Yak hemorrhage septicemia is an infectious bacterial disease”, there lost a sentence to describe infection diseases that threat the health of Yak.

Line 65-66, why vaccine could induce drug resistance against the bacteria? References should be provided.

Line 74-75, it’s not clear what is cp39 gene. Gene of P. multocida? “Poultry ompH gene” is incorrect, did it mean ompH gene of poultry P. multocida.

Line 83-85, this sentence could be deleted or it can be moved to other places. Otherwise, the logic is a bit confusing.

Line 119, the methods for constructing OmpH-deficient strains(ΔOmpH) should be described clearly.

Author Response

(The authors gave the same response as above.)

Round 2

Author Response

(The authors gave the same response as above.)

Reviewer 2 Report

The authors have revised the manuscript accordingy. I have no other comments.

Author Response

Thank you for your suggestions!

Best regards!